# Transperineal Prostate Biopsy Under Local Anaesthesia, Tolerability, and Functional Outcomes: A Prospective, Monocentric, and Single-Operator Study

**DOI:** 10.3390/jcm14124377

**Published:** 2025-06-19

**Authors:** Gilles Adans-Dester, Mathieu Bourguignon, Guillaume Krings

**Affiliations:** 1Department of Urology, CHU UCL Namur, 5530 Yvoir, Belgium; mathieu.bourguignon@chuuclnamur.uclouvain.be; 2Department of Urology, UCL Cliniques Universitaires Saint-Luc, 1200 Brussels, Belgium; guillaume.krings@saintluc.uclouvain.be

**Keywords:** prostate cancer, prostate fusion biopsy, transperineal biopsy, local anaesthesia, patient selection

## Abstract

**Background**: Prostate cancer (PCa) remains a major health concern worldwide, although improved screening and treatments have reduced its incidence and mortality. MRI-targeted biopsies, especially using MRI–ultrasound fusion, enhance detection of clinically significant prostate cancer (CsPCa) and reduce unnecessary procedures. Transperineal biopsies offer the same diagnostic performance and reduce the risk of infection while limiting the need for antibiotic prophylaxis. However, they tend to be more painful under local anaesthesia and require greater operator experience. **Methods**: This study prospectively assessed the tolerability and effectiveness of transperineal targeted biopsies under local anaesthesia in a monocentric cohort of 51 patients. **Results**: Immediate pre-biopsy anxiety showed a clinically significant association with pain experienced during biopsies, and greater expected pain resulted in greater experienced pain. Overall patient tolerability was high. Local anaesthesia provided procedural flexibility, reduced resource utilisation, was cost-effective, and did not compromise precision. **Conclusions**: The results support local anaesthesia as a viable option, offering precision, patient satisfaction, and reduced healthcare resource utilisation. These results emphasise the importance of personalising the choice of anaesthesia modality for transperineal prostate biopsies, tailoring it to the patient’s anxiety. Larger studies are required to confirm these findings and validate the observed trends.

## 1. Introduction

Prostate cancer (PCa) is the most frequent cancer in the western male population and the second worldwide. Incidence and mortality have been declining in most countries due to improved screening and treatments [1]. PCa benefits from individual early detection with Prostate Specific Antigen (PSA) testing and digital rectal examination (DRE). Early detection should begin at age 50, or at age 40–45 in men with a family history of PCa, African origins, or who are BRCA mutation carriers. The detection and frequency of follow-up should be individually decided in discussion with the patient, based on PSA levels and life expectancy scores [2]. In the case of abnormal PSA values and/or DRE, further diagnostic evaluation with MRI imaging of the prostate may be indicated. MRI has proven a good negative predictive value of 90.8% (CI 95% 88.12–93.1%) for clinically significant PCa (CsPCa) (≥ISUP 2) detection [3]. However, the specificity is only 0.37 [4], which means that in the case of abnormal MRI findings, the patient will be offered prostate biopsies. Pre-biopsy MRI has been shown to be crucial, reducing unnecessary biopsies in 49% of patients [5] and improving the detection rate of CsPCa with MRI–ultrasound fusion biopsies [4,6]. The fusion method (cognitive vs. imaged-based) has shown only a trend in favour of image-based fusion, but without statistical significance in terms of cancer detection [7]. The biopsy pattern in biopsy-naive patients should be both targeted and systematic [4]. However, recent studies suggest limiting biopsies to targeted and side-specific biopsies, as most of the CsPCa will be found in a radius of 1 cm around the MRI index lesion [8]. Nevertheless, recent studies still emphasise the importance of both targeted and systematic biopsies [9], with 30% of CsPCa found outside this region + 1 cm radius and 15% in the contralateral lobe [10].

Another recent shift has occurred in the way prostate biopsies are accessed, with the trend now favouring transperineal prostate biopsies over the traditional transrectal approach [2]. Transperineal biopsies reduce the risk of sepsis by almost 10 times [11], and they carry no additional infectious risk when performed without antibiotic prophylaxis compared to transrectal biopsies with prophylaxis. This is particularly important in the context of increasing antibiotic resistance and ongoing efforts to minimise antibiotic use [12]. Regarding cancer detection, the diagnostic accuracy remains the same [13]. This can be performed under local anaesthesia, even if the associated pain is higher for transperineal than for transrectal biopsies [13,14]. Severe anxiety was identified as the most significant factor contributing to pain. Age and longer operative time also influenced the level of pain experienced during transperineal prostate biopsies [15]. Despite its advantages, transperineal access is not yet widely adopted within our medical community. The need for equipment modifications, financial investment, and a learning curve of approximately 70 cases to achieve operative times and ease of use comparable to the transrectal approach under local anaesthesia may discourage some practitioners from transitioning to the transperineal technique [16].

Performing prostate biopsies under local anaesthesia offers more flexibility, as it requires less human power and less medical infrastructure. It can also reduce the use of costly medico-anaesthetic infrastructure used in the general anaesthesia/sedation setting. It also minimises the procedure’s impact on the patient, who can quickly return to normal activities.

This study aims to assess the tolerability of this technique, which has been performed in the way described in this study for nearly 2 years in our department, but it has not been formally assessed yet in terms of patient selection and objectively reported tolerability.

## 2. Materials and Methods

We prospectively assessed the tolerability and operational effectiveness of transperineal MRI-targeted biopsies under pure local anaesthesia with a prospective, monocentric, mono-operator cohort of transperineal prostate biopsies. Patients were assigned for biopsies according to clinician decision following the “MRI pathway” [5] from our urological consultation in Godinne University Hospital (Yvoir, Belgium). This study was conducted between late January 2024 and late May 2024. The decision for local anaesthesia or sedation was made after discussion between the patient and the urologist, aiming to assess our current process of selection for biopsies. Out of 51 patients, 34 received local anaesthesia, whereas 17 opted for sedation. The prostate biopsies took place in a daycare clinic operative room, with the help of 1 nurse. If the biopsy was performed under sedation, an anaesthesiologist was present. Sedation was performed using propofol. No fasting was required for patients under local anaesthesia, whereas patients undergoing sedation had to observe a 6 h fast. All of our biopsies were performed transperineally, with image-based MRI-US elastic fusion using the Koelis Trinity^®^ platform (Koelis, Meylan, France), and following the MUSIC (Michigan Urological Surgery Improvement Collaborative) pattern. We targeted 3–4 biopsy cores for each MRI target lesion with a PIRADS score ≥4 or a PIRADS score of 3 with PSA density > 0.15 ng/mL/cc. In case of a mix of PIRADS 4–5 and 3 lesions, only PIRADS 4–5 were targeted. A preoperative urine analysis was performed to rule out bacteriuria. No antibiotic prophylaxis was used based on recent RCT [17] and local experience. The patient was placed in the lithotomy position and asked to self-retract the scrotum with the hands. We used cutaneous disinfection with a povidone–iodine solution and performed local anaesthesia with 20 mL Linisol 2% diluted with 20 mL NaCl 0.9%. We first performed a superficial anaesthesia with 20 mL of anaesthetic solution with a 21 G needle in a fan-shaped pattern from the median raphe down to each side to the anal margin (Figure 1). After the superficial anaesthesia, we performed MRI contouring on the Koelis Trinity^®^ software Promap 4.3.0. Then we performed prostatic/deep anaesthesia under transrectal ultrasound control with 10 mL of anaesthetic solution for each side, targeting the posterior–apical plane with the outer shaft of an 18 G biopsy needle (Figure 2). Once both superficial and prostatic anaesthesia were performed, transrectal ultrasound image acquisition, contouring, and image fusion were performed. After a short waiting period, prostate biopsies were performed using a 22 cm 18 G coaxial needle and a BD Magnum™ biopsy gun (Becton, Dickinson and Company, Franklin Lakes, NJ, USA).

For the 2 branches of the study, we asked the patient to complete a questionnaire electronically just before the biopsies, immediately after the biopsies, and 20 days after the procedure. We assessed the global anxiety with the State-Trait Anxiety Inventory (STAI-Y) [18]. We assessed the pre-interventional anxiety with the Amsterdam Preoperative Anxiety and Information Scale (APAIS) [19]. Pain was evaluated with a visual analogue scale (VAS) from 0 to 10 (10 being the most painful). The correlation between STAI-Y scores and anaesthesia choice was analysed with a T-test. We also searched for a correlation between pre-biopsy anxiety (APAIS) and pain during the biopsies. A correlation analysis was subsequently performed to assess the relationship between expected pain prior to the biopsies and the pain reported during the procedure. We also assessed the impact on lower urinary tract symptoms with the International Prostate Symptom Score (IPSS), along with a post-biopsy social and clinical evaluation (haematuria, haemospermia, sexual activity, social limitation) 20 days after the procedure. We asked the patient if they had experienced any limitation in their sexual activity or if the procedure had a psycho-social impact on their life with an easy binary question (Yes/No). Secondarily, we also performed a qualitative results analysis of the biopsies, assessing the percentage of biopsies with prostatic tissue and the percentage of targeted biopsies within the target. The duration of the procedure was also taken into account. We did not analyse the oncological results, as the size of the cohort and the control group were too small.

Statistical analysis was performed using JASP software (Version 0.19.3).

## 3. Results

A total of 51 patients were enrolled consecutively. The patients’ demographics are summarised in Table 1. The median age was 67 years (IQR 9.5 years; Q1 = 61.5, Q3 = 71), with a median PSA of 7 ng/mL (IQR: 4.75 ng/mL; Q1 = 5.5, Q3 = 10.25). The median size of the index lesion was 13 mm (IQR 8–20 mm). PIRADS classification was as follows: 35% PIRADS-4 (18/51), 20% PIRADS-5 (10/51), 16% PIRADS-3 (8/51), and 29% with no PIRADS lesion (15/51). A total of 29% of the lesions were from anterior localisation. Of the 51 patients, 34 underwent biopsies under pure local anaesthesia and 17 under sedation. The mean procedure time was 27.4 min under local anaesthesia versus 29 min under sedation. No patients experienced post-biopsy infection or retention, and all were discharged the same day. The median expected pain evaluation before biopsies using the VAS was 2 (IQR: 1.75–4), whereas the median pain evaluation during biopsies with local anaesthesia was 3 (IQR: 2–4). The median APAIS score was 8 in both groups, with a known cut-off of above 11 for major risk of per-procedural anxiety [19]. The median STAI-Y score was 32.5/80 (IQR: 29–36) and 39/80 (IQR: 34–40), with higher STAI-Y scores indicating higher global anxiety.

We performed a *T*-Test to investigate whether the choice of anaesthesia, determined by the surgeon and the patient, was influenced by the global anxiety of the patient (STAI-Y). This showed non-statistically significant results that could tend towards preferring sedation when basal anxiety was higher, with a T-Score value of −1.55 (*p*-value = 0.127, CI 95% [−7.02; 1.02]). A significant correlation was found between immediate pre-biopsy anxiety (APAIS) and pain during the biopsies (VAS) (Pearson correlation test with a coefficient 0.485, *p*-value = 0.0057, CI 95% [0.158; 0.716]). This indicates that pre-biopsy anxiety is significantly associated with pain during the procedure. However, the correlation between per-biopsy anxiety and pain during biopsy was not statistically significant (Pearson correlation test comparing per biopsy anxiety (STAI-Y) and VAS = 0.149, *p*-value = 0.424 CI 95% [−0.217; 0.478). Regarding the pain itself, there was no significant difference between expected pain and experienced pain (expressed in VAS) (T-score value = −1.19, *p*-value = 0.24 CI 95% [−0.34; 1.31]). However, when examining the relationship between expected pain and experienced pain using a Pearson correlation, we found a positive association: higher expected pain was correlated with higher experienced pain (r = 0.498, *p* = 0.0044, 95% CI [0.174; 0.724]).

The mean duration of haematuria was 3 days (range 0–20 days). At 20 days of evaluation, haemospermia was still present in 26% of the patients. Only four patients reported social limitations, two from each group. A total of 12 patients reported limitations in sexual activity, 4 in the sedation group and 8 in the local anaesthesia group. Regarding IPSS, the mean pre-biopsy IPSS was 8.7 and the mean IPSS at 20 days post procedure was 8.2. In response to the open question “would you do it again under local anaesthesia”, 76% of the patients with local anaesthesia answered “yes”, where 78% of these patients would also recommend it to a friend. The precision of biopsy targeting showed 89.7% of cores within the target, with 88.3% for local anaesthesia and 91.6% for sedation. In both groups, all targeted cores were within 1 cm of the target.

## 4. Discussion

Our results confirm that transperineal prostate fusion biopsies under local anaesthesia are well tolerated, as 76% of the patients would choose it again, and 78% would recommend it. Our study showed a correlation between immediate pre-biopsy anxiety and pain during the biopsies, as well as a correlation between a high expected pain and a higher experienced pain. This aligns with other studies [20]. We also observed a non-significant tendency towards greater pain with higher global anxiety (STAI-Y) that is not supported by statistical significance. If larger cohorts confirm this tendency, anxiety scores (STAI-Y) could help guide the choice of anaesthetic modalities that we can offer to the patient.

To minimise pre-procedural anxiety, techniques such as the use of music or virtual reality (VR) during the procedure could be useful. These techniques have been studied and had a positive effect on patients’ anxiety and pain [20,21]. Although we did not implement VR in this study, preliminary experience with it in our department outside of this study showed positive effects on patient relaxation. We tried the use of virtual reality (VR) headsets, outside of this study, with subjectively positive results on patient relaxation.

The transperineal approach under local anaesthesia provides a procedure of similar precision compared to general anaesthesia. This has been shown in different studies [13], and our limited data in this study show the same tendency. The impact on lower urinary tract symptoms was also low, and no major difference in IPSS between the two groups was seen. Oncological outcomes were not evaluated due to the small cohort size.

Proceeding under local anaesthesia, in addition to offering advantages for the patient, also benefits the practitioner by easing scheduling and reducing reliance on anaesthesiologists and operative rooms. Furthermore, it allows patients to resume normal activities more quickly, especially for those with professional and social commitments. Finally, from a more global and societal perspective, proceeding under local anaesthesia reduces the overall cost of the procedure.

Our study is limited by the fact that there was no randomisation performed. This was due to our aim to prospectively assess our department’s current procedure. This is the reason why we kept our current selection process, which was based on patient–doctor discussions. These discussions were mainly focused on the anxiety of the patient without using an objective scale. The selection criteria for sedation were quite high, as it was mainly performed for patients who declared themselves unable to overcome the stress of a local anaesthesia procedure. Another limitation could be the single-operator design. The operator (GAD) was considered as experienced, having performed 200+ transperineal prostate biopsies before starting the study. Finally, the small size of our cohort is certainly another limitation of our study, and it surely limits the generalisability of our results.

## 5. Conclusions

Transperineal prostate biopsies under local anaesthesia are feasible, well tolerated, and efficient. Given that both higher pre-biopsy anxiety and greater anticipated pain are significantly associated with increased pain during transperineal prostate biopsies under local anaesthesia, this finding supports the implementation of anxiety-reduction strategies and enhanced patient education prior to the procedure, or alternatively, the consideration of sedation for patients with high anticipatory distress. The anxiety scores of patients (STAI-Y) could help refine the assignment of anaesthesia modalities if the observed tendency we observed is confirmed in larger studies with statistical significance. Minimising pre-procedural stress and expected pain is important, as it correlates with reduced procedural pain. This technique provides more flexibility for urologists while lowering the socio-professional burden for patients and reducing costs. Further studies with larger cohorts are necessary to confirm the findings and trends observed in this monocentric study.

## Figures and Tables

**Figure 1 jcm-14-04377-f001:**
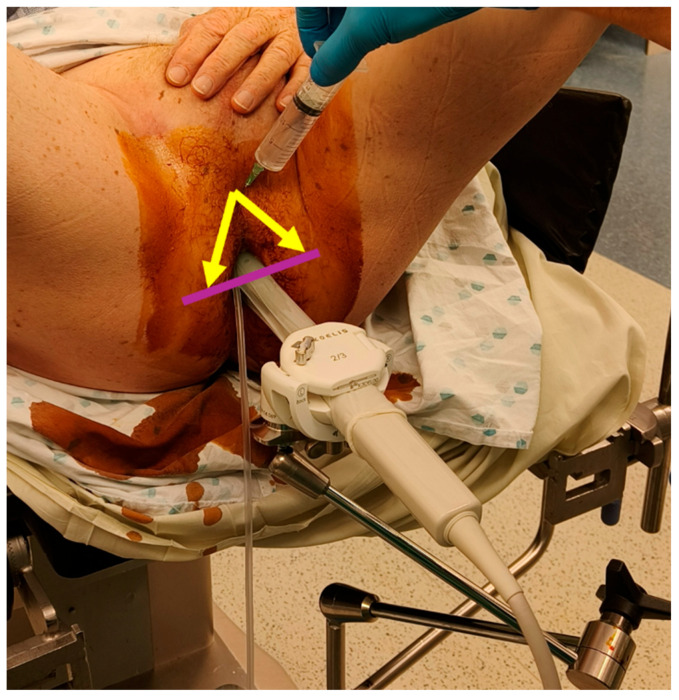
Superficial local anaesthesia in “fan shape” (arrows and line).

**Figure 2 jcm-14-04377-f002:**
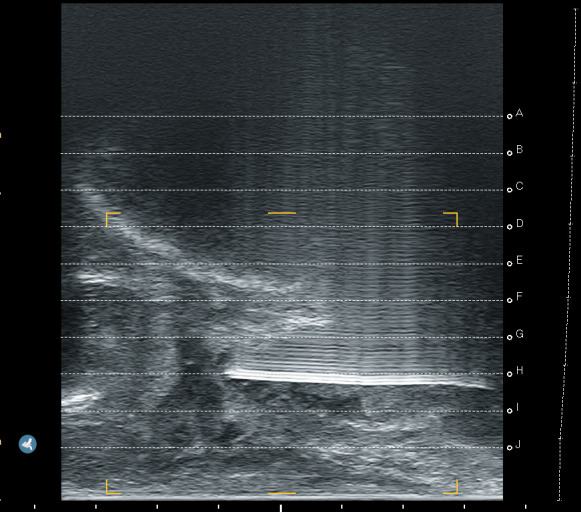
Prostatic anaesthesia.

**Table 1 jcm-14-04377-t001:** Population data.

	Local (*n* = 34)	Sedation (*n* = 17)
Median age (years)	66 (IQR: 59.4–70)	67 (IQR: 63.0–71.0)
Median PSA (ng/mL)	7 (IQR: 6–10)	6.7 (IQR: 4.8–10.5)
Mean PSA (ng/mL)	15 (1.3–124)	11.4 (0.5–15.3)
PIRADS 0–2	26% (9/34)	35% (6/17)
PIRADS 3	21% (7/34)	6% (1/17)
PIRADS 4	29% (10/34)	47% (8/17)
PIRADS 5	24% (8/34)	12% (2/17)
Median index lesion size (diameter)	13 mm (IQR: 8–20)	9.5 mm (IQR: 6.3–13.7)
Anterior lesion	27% (9/34)	18% (3/17)
Mean number of cores	16.5 (14–22)	16.5 (14–22)
ISUP ≥ 2 at biopsies	47% (16/34)	35% (6/17)
Median STAI-Y score (anxiety trait)	32.5/80 (IQR: 29–36)	39/80 (IQR: 34–40)
Median APAIS score	8 /20(IQR: 6–13.75)	8 /20(IQR: 4–11)
Median pain expected (EVA)	2 (IQR: 1.75–4)	Not Applicable
Median pain during biopsies (EVA)	3 (IQR: 2–4)	Not Applicable
Operative time	27.4 min	29 min

## Data Availability

Data available on request to the author for personal data protection (small cohorts within limited time frames and geographic area).

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
