# Peer review of "Transperineal Prostate Biopsy Under Local Anaesthesia, Tolerability, and Functional Outcomes: A Prospective, Monocentric, and Single-Operator Study"

_jcm, 2025, doi:10.3390/jcm14124377_

Round 1
Reviewer 1 Report
Comments and Suggestions for Authors
The authors present a prospective, single-operator, monocentric study evaluating tolerability and functional outcomes of transperineal MRI-targeted prostate biopsies under local anaesthesia. The manuscript is well written and addresses a clinical point in prostate biopsy approach.
- The use of terms like “trend” for non-significant findings should be applied cautiously. The authors could better contextualize these trends within the limits of the small sample size.
- P-values are reported, but confidence intervals for key findings (e.g., pain scores, correlations) would add value.
- The manuscript states 26% of patients reported hemospermia at 20 days, but no comparison with baseline data or literature is provided. Is this consistent with previous studies?
Author Response
Dear Reviewers,
We would like to thank you for your thoughtful, constructive, and insightful comments. Scientific writing is a complex discipline that requires both experience and deep understanding. Thanks to your feedback, we were able to improve not only the quality of our manuscript, but also our own knowledge and experience. We sincerely hope that our answers and revisions adequately address your questions and concerns.
We also slightly expanded the abstract, introduction and methods sections, as requested by the editor, in addition to incorporating your suggestions. Each comment was addressed separately in the answer letter and also directly in the revised manuscript using track changes for clarity. We also provide a revised version of the manuscript with only the updated sections and changes highlighted.
Kind Regards,
Gilles A-D, Guillaume K, Mathieu B.
Reviewer 1 :
- The use of terms like “trend” for non-significant findings should be applied cautiously. The authors could better contextualize these trends within the limits of the small sample size.
We would like to apologize for the excessive use of the term “trend,” which may reflect an unfortunate linguistic translation, as it does not carry the same interpretative weight in the vernacular use of French. We have replaced it with more appropriate alternatives such as:
- Page 4 Line 156-158: This showed a non-statistically significant results that could tend towards preferring sedation when basal anxiety was higher, with a T-Score value of -1.55 (p-value= 0.127, CI 95% [-7.02; 1.02]).
- Page 5 Line 185-187: We also observed a non-significant tendency towards greater pain with higher global anxiety (STAI-Y) that is not supported with statistical significance
- Page 5 Line 187-189: If larger cohorts confirm this tendency, anxiety scores (STAI-Y) could help guide the choice of anaesthetic modalities that we can offer to the patient.
- Page 5 Line 223-225: Anxiety scores of the patient (STAI-Y) could help refine the assignment of anaesthesia modalities if the observed tendency we observed is confirmed in larger studies with statistical significance.
- P-values are reported, but confidence intervals for key findings (e.g., pain scores, correlations) would add value.
All results are now reported with their corresponding p-values and 95% confidence intervals. We apologize for not providing this information earlier.
- The manuscript states 26% of patients reported hemospermia at 20 days, but no comparison with baseline data or literature is provided. Is this consistent with previous studies?
The section on hemospermia has been further developed and contextualized with references from the literature.
Page 4 Line 170-176: At the 20 days evaluation, hemospermia was still present in 26% (13/51) of the patients and 47% (24/51) experienced hemospermia. The literature describes hemospermia as the most common complication in prostate transperineal biopsy ranging from 30.4% [20]to 6.1 % [21] but also up to 93% for transrectal biopsies[22]. The observed discrepancy may reflect a reporting bias. Nonetheless, when hemospermia was more frequently reported, it was a source of concern for about one-fourth of the patients[22]
- Lines 36-37 "MRI has proven sensitivity and specificity of 0.91 and 0.37, respectively". Urologists use MRI to rule out high grade PCa, please provide only the negative predictive value, which is around 90%. Also provide a more general view, not a specific one "MRI has negative predictive value between xx% and yy%".
Thank you for highlighting this point. Indeed, the most relevant information for both clinicians and patients is the negative predictive value (NPV), and it should therefore be explicitly stated. For the sake of completeness, we also included specificity, as its low value underscores the need to proceed with biopsies.
We changed for:
Page 2 Line 56-49“MRI has proven a good negative predictive value of 90.8% (CI 95% 88.12-93.1%) for clinically significant PCa (CsPCa) (≥ISUP 2) detection [3]. However, the specificity is only of 0.37 [4]. Which means that in case of abnormal MRI findings, the patient will be offered prostate biopsies.”
- General comment about statistics: please carefully read all major urology journal statistical guidelines:
-https://pubmed.ncbi.nlm.nih.gov/30580902/
-https://pubmed.ncbi.nlm.nih.gov/32451178/
-https://pubmed.ncbi.nlm.nih.gov/37286459/
We have taken your comment seriously, and it has prompted substantial revisions to the data presentation, which was initially lacking in clarity and completeness.
We also conducted a correlation analysis between expected and experienced pain, which had not been considered in the initial version, and the results proved to be statistically significant.
Reviewer 2 Report
Comments and Suggestions for Authors
Lines 36-37 "MRI has proven sensitivity and specificity of 0.91 and 0.37, respectively". Urologists use MRI to rule out high grade PCa, please provide only the negative predictive value, which is around 90%. Also provide a more general view, not a specific one "MRI has negative predictive value between xx% and yy%".
General comment about statistics: please carefully read all major urology journal statistical guidelines:
-https://pubmed.ncbi.nlm.nih.gov/30580902/
-https://pubmed.ncbi.nlm.nih.gov/32451178/
-https://pubmed.ncbi.nlm.nih.gov/37286459/
There are some issues related to decimal figures, take as example "median PSA of 14.16 107 ng/ml", it is highly unlikely that there is a meaningful difference between PSA 14.16 and 14.17, please consider reporting only one decimal figure for PSA (or even no decimals). Authors also report ranges instead of interquartile ranges, the latter are preferred to ranges. Same for PI-RADS score distrubution, please provide number and percent, not only percent. A mean duration of hematuria of 2.75 days suggests that the authors can split the day in minutes, please avoid using decimals here, just say 2 days.
The authors use the word "trend" a lot when describing statistical association, which is problematic. I encourage to read Eur Urol stats guideline 3.2 "P values just above 5% are not a trend"
Line 164 "Our study is limited by the fact that there was no real randomization done". Please remove "real", a randomization is either a randomization or not, it cannot be real or false.
Line 176 "Better patient selection is key to personalized and patient-tailored medicine" this is a too general statement. Please consider the Eur Urol guidelines on causality in observational research: Authors must draw conclusions and draw implications for clinical practice or further research. You found a correlation between anxiety prior to biopsy and pain, which can be a good indication of procedures aimed at reducing stress prior to LATP. Such step is nicely performd at the end of the conclusion.
Table 1 has insufficient data and imprecise annotation, please refer to Eur Urol statistics guidelines for tables and figures and consider adding PSA density and biopsy results, as well as number of cores taken and number of positive cores at least.
Author Response
Dear Reviewers,
We would like to thank you for your thoughtful, constructive, and insightful comments. Scientific writing is a complex discipline that requires both experience and deep understanding. Thanks to your feedback, we were able to improve not only the quality of our manuscript, but also our own knowledge and experience. We sincerely hope that our answers and revisions adequately address your questions and concerns.
We also slightly expanded the abstract, introduction and methods sections, as requested by the editor, in addition to incorporating your suggestions. Each comment was addressed separately in the answer letter and also directly in the revised manuscript using track changes for clarity. We also provide a revised version of the manuscript with only the updated sections and changes highlighted.
Kind Regards,
Gilles A-D, Guillaume K, Mathieu B.
Reviewer 2 :
- There are some issues related to decimal figures, take as example "median PSA of 14.16 107 ng/ml", it is highly unlikely that there is a meaningful difference between PSA 14.16 and 14.17, please consider reporting only one decimal figure for PSA (or even no decimals).
We fully agree that reporting more than one decimal place lacks meaningful precision. As it was actually the mean, we updated the value for the real median which is 7ng/ml (IQR: 4.75ng/ml; Q1= 5.5, Q3= 10.25). Page 4 line 142 - Authors also report ranges instead of interquartile ranges, the latter are preferred to ranges. Same for PI-RADS score distribution, please provide number and percent, not only percent.
We have replaced all means with medians and ranges with interquartile ranges (IQR). In addition to reporting percentages in the population data table, we have now included the absolute numbers as well, both in the table and in the main text for completeness.
- A mean duration of hematuria of 2.75 days suggests that the authors can split the day in minutes, please avoid using decimals here, just say 2 days.
We fully agree with the comment and have adjusted the value to a more clinically meaningful one, rounding it from 2.75 days to 3 days. Page 4 line 174
- The authors use the word "trend" a lot when describing statistical association, which is problematic. I encourage to read Eur Urol stats guideline 3.2 "P values just above 5% are not a trend"
We would like to apologize for the excessive use of the term “trend,” which may reflect an unfortunate linguistic translation, as it does not carry the same interpretative weight in the vernacular use of French. We have replaced it with more appropriate alternatives such as:
- Page 4 Line 156-158: This showed a non-statistically significant results that could tend towards preferring sedation when basal anxiety was higher, with a T-Score value of -1.55 (p-value= 0.127, CI 95% [-7.02; 1.02]).
- Page 5 Line 185-187: We also observed a non-significant tendency towards greater pain with higher global anxiety (STAI-Y) that is not supported with statistical significance
- Page 5 Line 187-189: If larger cohorts confirm this tendency, anxiety scores (STAI-Y) could help guide the choice of anaesthetic modalities that we can offer to the patient.
- Page 5 Line 223-225: Anxiety scores of the patient (STAI-Y) could help refine the assignment of anaesthesia modalities if the observed tendency we observed is confirmed in larger studies with statistical significance.
- Line 164 "Our study is limited by the fact that there was no real randomization done". Please remove "real", a randomization is either a randomization or not, it cannot be real or false.
We agree with this comment and its rationale. We have changed “real randomisation” to simply “randomisation.” Page 5 line 211
- Line 176 "Better patient selection is key to personalized and patient-tailored medicine" this is a too general statement. Please consider the Eur Urol guidelines on causality in observational research: Authors must draw conclusions and draw implications for clinical practice or further research. You found a correlation between anxiety prior to biopsy and pain, which can be a good indication of procedures aimed at reducing stress prior to LATP. Such step is nicely performd at the end of the conclusion.
We reformulated to :
Page 5 Line 223-227: Given that both higher pre-biopsy anxiety and greater anticipated pain are significantly associated with increased pain during transperineal prostate biopsies under local anesthesia, this finding supports the implementation of anxiety-reduction strategies and enhanced patient education prior to the procedure, or alternatively, the consideration of sedation for patients with high anticipatory distress.
- Table 1 has insufficient data and imprecise annotation, please refer to Eur Urol statistics guidelines for tables and figures and consider adding PSA density and biopsy results, as well as number of cores taken and number of positive cores at least.
We have revised Table 1 to include more detailed data and have replaced means with medians and ranges with interquartile ranges (IQR), in accordance with the guidelines. We chose, however, to retain the mean PSA value in order to reflect the presence of extreme outliers.
Unfortunately, some of the suggested data were not collected in our study, as we intentionally chose not to evaluate oncological outcomes (e.g., number of positive cores, PSA density, side affected, concordance with MRI). These variables will be addressed in future work, ideally with a larger cohort.
That being said, if you consider these data essential, we are able to retrieve them—though this would require additional time. Given the short review deadline (June 1st) and our current clinical workload, we regret that we are unable to provide this information within the requested timeframe.